# Efficient Hole Transfer from a Twisted Perylenediimide Acceptor to a Conjugated Polymer in Organic Bulk-Heterojunction Solar Cells

**DOI:** 10.3390/ma16020737

**Published:** 2023-01-12

**Authors:** Hyojung Cha

**Affiliations:** Department of Hydrogen and Renewable Energy, Kyungpook National University, Daegu 41566, Republic of Korea; hcha@knu.ac.kr

**Keywords:** organic solar cells, non-fullerene acceptor, hole transfer, perylenediimide, transient absorption spectroscopy

## Abstract

Non-fullerene acceptors have recently attracted tremendous interest due to their potential as alternatives to fullerene derivatives in bulk-heterojunction solar cells. Nevertheless, physical understanding of charge carrier generation and transfer mechanism that occurred at the interface between the non-fullerene molecule and donor polymer is still behind their enhanced photovoltaic performance. Here we report examples of a non-planar perylene dimer (TP) as an electron acceptor and achieve a power conversion efficiency of 6.29% in a fullerene-free solar cell. Photoluminescence (PL) measurements show high quenching efficiency driven by the excitons of both conjugated polymer and TP molecule, respectively, indicating efficient electron and hole transfer, which can support a highly intermixed phase of blends measured by atomic force microscopy (AFM) and grazing incident wide-angle X-ray diffraction (GIWAXS). Femtosecond transient absorption spectroscopy (fs-TAS) reveals that the fast exciton dissociation process from TP molecule to donor polymer contributes to additionally increasing current density, leading to stronger incident photon to current efficiency in the visible region.

## 1. Introduction

Non-fullerene acceptors have advanced considerably over recent years and have achieved power conversion efficiencies (PCEs) of over 18%, surpassing the photovoltaic performance of fullerene derivatives-based solar cells by designing the molecular structure of electron donating/accepting materials, optimizing phase separation morphology of photoactive layers, developing interfacial layers and designing a new type of device configurations [1,2,3,4,5,6,7,8,9,10,11]. The high-performance non-fullerene acceptors possess a strong light absorption ability in the visible and near-infrared (NIR) regions and tuneable electronic properties to increase open circuit voltage (*V*_OC_) [12,13]. Accordingly, the molecular design, synthesis, and device testing of new concepts for non-fullerene acceptors have been in progress with enormous interest.

Among the recently reported novel non-fullerene electron-accepting small molecules, perylenediimides (PDIs) have been making noticeable progress due to their high thermal, chemical, and photochemical stabilities, light absorption in the visible region, strong electron affinity, and easy functionalization of the aromatic core or imide positions for optoelectronic and self-assembling properties [12,13]. The introduction of the 2D- or 3D-twisted concept in the PDI-type electron acceptors has been investigated to weaken the strong intermolecular interaction of PDI units and avoid large crystalline aggregate domains. The twisted perylenediimide (TP) molecules have been designed by linking at the imide positions or bay positions without compromising their charge transport properties [14,15,16,17]. The TP blend solar cells incorporating PBDTTT-C-T have exhibited an encouraging PCE of 3.2% with a *V*_OC_ of 0.77 V, a short circuit current (*J*_SC_) of 9.0 mA cm^−2^ and a fill factor (FF) of 0.46 [14,15]. The twisted single bond linked at the imide position of the PDI molecule attributed to minimizing the electronic repulsion between the oxygen atoms, which significantly weakened the intermolecular aggregation effect [12,13]. This molecular structural design concept motivated the development of new strategies to produce dimeric or tetrameric PDIs [14,15]. Thus, these 2D- or 3D-twisted PDI molecules have been systematically designed and achieved up to 9.5% with a spiro-bifluorene bridge at the bay position between two PDIs [18]. The relatively high performance of the PDI-based bulk-heterojunction (BHJ) solar cells was mainly ascribed to a combination of factors with electron-donating materials, including complementary light absorption, an adjusted energetic difference for higher open circuit voltage, and optimal nanoscale phase separation morphology [18]. However, the photo-physical understanding of such non-fullerene-based active layers still lags behind that of fullerene-based active layers incorporating conjugated polymer. Especially the charge carrier generation mechanism in electron-accepting materials, which is strongly related to exciton dissociation from electron acceptor to electron-donating materials, is less studied in fullerene-free bulk-heterojunction solar cells.

In this paper, we demonstrated that fullerene-free BHJ solar cells incorporating a TP as an electron acceptor achieved PCEs of 6.29%, with high *J*_SC_ of 13.94 mA cm^−2^ and *V*_OC_ of 0.84 V, compared the nanoscales morphologies of PffBT4T-2OD:TP blend films at different composition ratios, and correlated these with exciton generation, exciton dissociation, and charge carrier recombination dynamics. We directly investigate the motion of charge carriers in PffBT4T-2OD:TP between the different phases, thus clearly establishing their interplay and determining their role in defining the photo-physics of free-charge generation in TP acceptors. Importantly, we find that the ability of photo-generated electron–hole pairs to dissociate is already determined by the acceptor phase at ultrafast times, while the exciton generation and separation of the charge pairs in conjugated polymers can be identical in the blends of different ratios.

## 2. Materials and Methods

### 2.1. Device Fabrication and J-V Characterization

ITO-coated glass substrate (PsioTec Ltd., Berkhamsted, UK, ≈15 Ω square^−1^) was cleaned by sonication in detergent, DI water, acetone, and isopropanol for 15 min for each step. A ZnO film (40 nm) as an electron transport layer was prepared by spin-coating at 4000 rpm from a ZnO precursor (diethyl zinc) solution after oxygen plasma for 10 min at 100 W. Active layer solutions (donor concentration 10 mg mL^−1^) were prepared in chlorobenzene(CB):o-dichlorobenzene (o-DCB) (1:1 volume ratio) with 3 vol% of diiodooctane (DIO). Active layers were spin-coated on the preheated ITO/ZnO film substrate, where MoO_3_ (10 nm)/Ag (100 nm) electrodes were deposited by thermal evaporation under vacuum.

Device area was 0.045 cm^2^. *J*–*V* characteristics were measured using a Xenon lamp at AM1.5 solar illumination (Oriel Instruments) calibrated to a silicon reference cell with a Keithley 2400 source meter, correcting for spectral mismatch.

External quantum efficiency (EQE) was measured by Bentham IL1 with Bentham 605 stabilized current power supply with a 100 W tungsten halogen lamp coupled to a monochromator with computer-controlled stepper motor.

### 2.2. UV-Vis Absorption and Photoluminescence (PL) Spectroscopies

UV-Visible spectra of thin films on glass substrates were measured with a Lambda 25 spectrometer (Perkin Elmer, Buckinghamshire, UK). PL spectra were acquired with a Fluorolog-3 spectrofluorometer (Horiba Jobin Yvon, Kyoto, Japan).

### 2.3. AFM Characterization and GIWAXS Characterization

GIWAXS measurements were performed at beamline 3C at the Pohang Accelerator Laboratory (PAL) in South Korea. The 10 keV X-ray beam was incident at a grazing angle of 0.12–0.16°, selected to maximize the scattering intensity from the samples. The 2D scattered X-rays were detected using a Dectris Pilatus 2M photon counting detector (Switzerland).

AFM measurements were measured with a Scanning Probe Microscope-Dimension 3100 (Veeco, Plainview, NY, USA) in tapping mode.

### 2.4. Transient Absorption Spectroscopy

Femtosecond TAS was performed using a transient absorption spectrometer, HELIOS (Ultrafast systems, Sarasota, FL, USA), with a pulse train generated by an optical parametric amplifier, TOPAS (Light conversion). The spectrometer and the parametric amplifier were seeded with 800 nm, <100 femtosecond pulses at 1 kHz generated by a Solstice Ti:Sapphire regenerative amplifier (Newport Ltd., North Logan, UT, USA).

## 3. Results and Discussion

The BHJ cells using TP as an electron acceptor are based on the commercially available low bandgap conjugated polymer PffBT4T-2OD as an electron donor, as shown in Figure 1a, and fabricated with an inverted device configuration of glass/indium tin oxide (ITO)/zinc oxide (ZnO)/active layer/molybdenum trioxide (MoO_3_)/silver (Ag). The energy levels of the materials used in this work were determined by corresponding values reported in the literature (Figure 1b) [12,13]. The energetic difference (0.33 eV) between LUMO levels of donor and acceptor materials in the photoactive layer indicates favorable energy offsets for charge separation between PffBT4T-2OD and TP molecules. For devices based on the TP acceptor, we obtain device performance with the donor:acceptor blend layers at a different weight ratio of 1:1 and 1:2, spin-coated from an o-dichlorobenzene (o-DCB). Current density-voltage (*J*–*V*) curves for PffBT4T-2OD:TP blend solar cells are shown in Figure 1c. The solar cell parameters are listed in Table 1. The PffBT4T-2OD:TP (1:1) blend solar cell exhibits a *J*_SC_ of 13.94 mA cm^−2^ compared to 12.78 mA cm^−2^ for that of 1:2 (Figure 1c). Due to the low-lying HOMO level of PffBT4T-2OD, PffBT4T-2OD:TP blends show higher *V*_OC_ (0.84 V) compared to our previous work used PBDTTT-C-T (0.77 V) [13].

Figure 1d exhibits the UV-Visible absorption spectra and the external quantum efficiency (EQE) spectra for each device at different acceptor composition ratios. TP molecules absorb light strongly in the wavelength range from 400 to 600 nm with a maximum absorption peak of 547 nm, which makes complementary absorption spectra with PffBT4T-2OD (λ_max_ = 692 nm) an advantage for photon harvesting in the visible region. These devices show broad and strong photo responses from 400 to 700 nm in entire EQE spectra. In the wavelength range of 400–600 nm, the larger photo response strength of TP compared to PffBT4T-2OD can be attributed to a higher photoinduced hole transfer from TP excitons. Similarly, in the 600–700 nm ranges, the higher light absorption of PffBT4T-2OD should predominantly lead to electron transfer from PffBT4T-2OD excitons. The integrated *J*_SC_ values for PffBT4T-2OD:TP blends of weight ratio of 1:1 and 1:2 are 13.35 and 12.50 mA cm^−2^, respectively, which agree well with measured values within 5% for all the devices. However, the EQE spectrum for the PffBT4T-2OD:TP solar cell, including a blend of weight ratio of 1:1, shows a much higher increase at 450 nm due to the intense absorption band for TP. For this PffBT4T-2OD:TP solar cells, the highest EQE values are even beyond 70%. The important conclusion is that the photogenerated excitons from electron acceptors contribute greatly to the photocurrent in these solar cells.

To investigate the yield of exciton dissociation at the interface between PffBT4T-2OD and TP domains, we performed steady-state photoluminescence (PL) spectroscopy. For the PL data, excitation wavelengths of 600 nm for PffBT4T-2OD or 480 nm for TP were employed to achieve mainly selective excitation of the electron donor polymer or the electron acceptor small molecule, respectively. The PL spectra of neat PffBT4T-2OD and TP films indicate emission maximal emission wavelength at 742 nm and 650 nm, respectively. Compared to the PL of neat films, that of blend films indicates high exciton quenching efficiency (95~99%) of both the PffBT4T-2OD and TP emission and suggests efficient exciton dissociation between PffBT4T-2OD and TP driven by both PffBT4T-2OD and TP excitons (Appendix A). The PL quenching yield in TP-incorporating blends does not show any noticeable difference, which is attributed to the similar tendency of the intermixed phase and is supported by the results of both similar atomic force microscopy (AFM) images.

AFM was employed to investigate the surface morphology of the photoactive layers, including 1:1 and 1:2 ratio blend films coated on the ITO substrate under similar fabrication conditions as those of devices. The height images of all optimal blend films are shown in Figure 2a,b. The scans of 1 µm × 1 µm show the surface morphology of both blends with an average root mean square roughness of 2.50 nm. AFM height images in an optimized condition of PffBT4T-2OD:TP blends show that the device blends have similar features and comparatively similar smoothness. This implies that the highly intermixed phase of both PffBT4T-2OD and TP in the blends is achieved, which is consistent with the fact that PL data show high exciton quenching. To reveal the microstructure of PffBT4T-2OD:TP blend films prepared with different ratios of 1:1 and 1:2, respectively, we characterize the pure TP and both PffBT4T-2OD:TP blend films by grazing incidence wide-angle X-ray scattering (GIWAXS). The GIWAXS two-dimensional (2D) maps of the pure TP and both PffBT4T-2OD:TP blend films are shown in Figure 2c–e. We note that the change from blue to red represents the improvement. The TP film does not exhibit high-order lamellar stacking peaks, and the peak intensity is quite low. TP acceptor incorporation in both blends also diminishes the intermolecular orientation of PffBT4T-2OD. These GIWAXS results can support the results obtained from PL data and AFM images, which are in good agreement with the highly intermixed phase of both blends.

In a previous morphology study with respect to the blend, layers consisted of PffBT4T-2OD and TP molecules with different ratios of composition; we found that PffBT4T-2OD:TP blend layer exhibited high intermixing of the donor-acceptor phases due to the good miscibility of TP into the high crystalline conjugated polymers. However, the morphology of the PffBT4T-2OD-based blend cannot explain the performance difference between blends of weigh ratio of 1:1 and 1:2. For the further understanding of exciton generation in the relatively pure donor or acceptor phase and exciton dissociation efficiency at the interface between PffBT4T-2OD and acceptor materials, fs-TAS measurement is employed to establish the dual light-harvesting mechanism and quantify the charge transfer and recombination dynamics in both blends [19,20,21]. In the optical properties, it was seen that the TP electron acceptors could complement low-bandgap electron donors for efficient harvesting of solar radiation in a broad wavelength region. Appendix A compares transient absorption spectra for neat films of TP and PffBT4T-2OD on photoexcitation at 500 and 715 nm, respectively. To measure the exciton lifetime by fs-TAS, an excitation of 500 nm (Appendix A), corresponding to the dominant absorption of the acceptor molecule, was used to ensure selective excitation of the TP to study the charge generation dynamics from polymer excitons. The spectrum from neat TP excited at 500 nm features a ground-state bleaching peak at 550 nm and a broad excited state absorption peak at 767 nm, and this feature decays mono exponentially with an exciton lifetime of 581 ps at the fluence of 3.86 µJ cm^−2^. The longer singlet exciton lifetimes of TP can provide more efficient exciton diffusion prior to exciton dissociation. On the other hand, in the excitation at 715 nm (Appendix A), the spectrum from neat PffBT4T-2OD features a broad excited state absorption peak at 1250 nm, this feature decays with exciton lifetime of 359 ps at the fluence of 2.54 µJ cm^−2^, which is also relatively longer than that of other conjugated polymers. PffBT4T-2OD:TP blends (Appendix A) excited at 715 nm show the excited state absorption feature decays rapidly, compared to singlet exciton lifetime in the neat polymer film (359 ps), with time constants of 35 and 30 ps in 1:1 and 1:2 ratios, respectively, suggesting ultrafast electron transfer from photoexcited PffBT4T-2OD to the electron acceptor TP. The ultrafast decay of the excited state absorption peak at 1250 nm is accompanied by a new excited state absorption at 1100 nm that is assigned to polaron absorption in PffBT4T-2OD. Interestingly, regardless of different blend ratios, both films of weight ratios of 1:1 and 1:2 exhibit identical electron transfer dynamics from PffBT4T-2OD to TP.

We now turn to complementary light harvesting by TP at its peak absorption wavelength of 547 nm (see Figure 1d). As shown in Figure 3a,b, the excitation at 500 nm leads to ground-state bleaching peaks at 550 nm for the TP acceptor and at both 625 and 700 nm for the PffBT4T-2OD donor, and a simultaneous broad excited state absorption feature in the range of 725–1300 nm at the delay time of 213 fs. Due to the different load of acceptors in both blends, the different intensities of transient absorption in both blends at the delay time of 213 fs are observed at the ground state bleaching of 550 nm, while the ground state bleaching of PffBT4T-2OD at 625 and 700 nm show similar behavior. The excitation of the TP acceptor at 500nm leads to a rise in the bleaching of PffBT4T-2OD at 700 nm, which is assigned to ultrafast hole transfer from photoexcited TP to PffBT4T-2OD. Moreover, the bleaching signal of PffBT4T-2OD grows with a time constant of 3 ps for the blend of the ratio of 1:1 and 10 ps for the blend of the ratio of 1:2, respectively (Figure 4b). Simultaneously, the bleaching signal of the TP acceptor at 560 nm continuously decreases with a time constant of 30 ps for both blends (Figure 4a). This interpretation is supported by the growth of excited state absorption of polaron absorption in PffBT4T-2OD:TP blends at 1100 nm (Figure 4c). Therefore, higher yields of excited state absorption of singlet exciton probed at 1250 nm as well as long-lived polaron probed at 1100 nm were found in the blend film with a weight ratio of 1:1 compared to that of 1:2 (Figure 4d). The transient absorption spectra and dynamics presented that photoexcitation of either the donor or the acceptor contributes to the charge-carrier generation and charge transfer. Ultrafast exciton generation from these acceptors in the blend film compensates for the photoexcitation of the donor, resulting in a strong photo response over the visible light region in EQE.

Microsecond transient absorption studies of blend films were carried out under low-intensity excitation conditions of 3.5 µJ cm^−2^ (Appendix A) [19]. The transient absorption of a PffBT4T-2OD:TP blend film at 0 to 10 µs also indicates enhanced broad near-infrared absorption in the range 750–1000 nm assigned to the formation of PffBT4T-2OD positive polarons. The negative magnitude of the transient absorption signal for <750 nm arises from the ground state bleaching of the PffBT4T-2OD ground state. Appendix A shows a comparison of this PBDTTT-CT polaron signal between PffBT4T-2OD:TP (1:1) films under nitrogen or oxygen, employing matched excitation densities (3.5 µJ cm^−2^). The transient signals exhibited oxygen-independent decay dynamics, indicative of their assignment to polarons rather than triplet absorption.

## 4. Conclusions

Fullerene-free bulk-heterojunction solar cells exhibited higher power conversion efficiency with a new combination of TP-based blends consisting of PffBT4T-2OD. High PL quenching efficiency of both PffBT4T-2OD and TP emissions revealed efficient charge transfer at the electron-acceptor interface with a highly intermixed domain of PffBT4T-2OD and TP. The EQE measurements demonstrated that current could be generated from both the PffBT4T-2OD and TP excitons. By measuring fs-TAS, the blend of mass ratio of 1:1 showed identical electron transfer dynamics and comparable yield of long-lived polaron from the excitons of PffBT4T-2OD, while hole transfer from polymer to TP occurred more efficient transient absorption and faster exciton dissociation, as compared to that of 1:2. This efficient charge generation and transfer from TP acceptor give a significant effect to the current generation with high photoresponsivity up to 70% in visible regions. Consequently, the TP molecule has morphological and photophysical potentials as a promising acceptor for efficient, fullerene-free BHJ solar cells.

## Figures and Tables

**Figure 1 materials-16-00737-f001:**
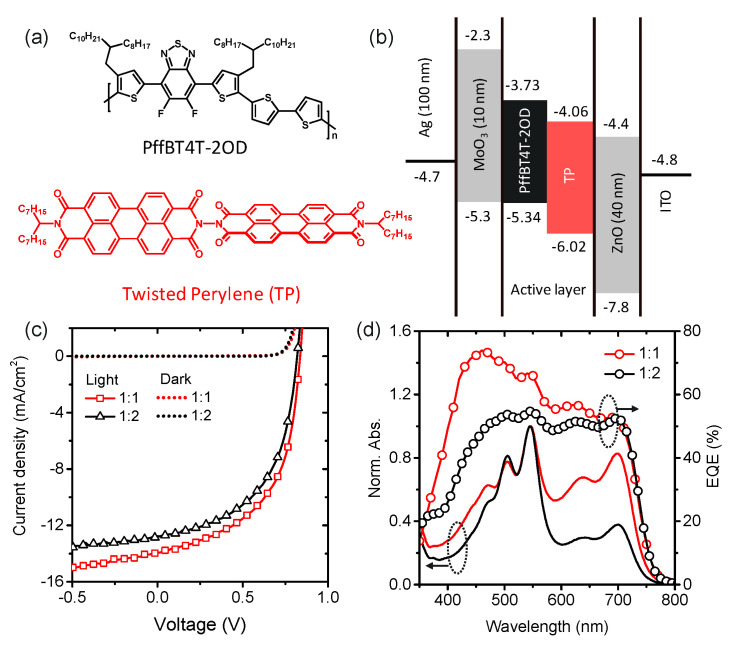
(**a**) Structure of PffBT4T-2OD and twisted perylenediimide (TP). (**b**) Energy levels of the materials in the inverted device structure used in this study. (**c**) *J*–*V* characteristics of PffBT4T-2OD:TP devices with different compositional ratios of 1:1 (red) and 1:2 (black) under illumination of an AM 1.5G solar simulator (100 mW cm^−2^). (**d**) EQE and absorption spectra of PffBT4T-2OD:TP at different weight ratios of 1:1 (red) and 1:2 (black).

**Figure 2 materials-16-00737-f002:**
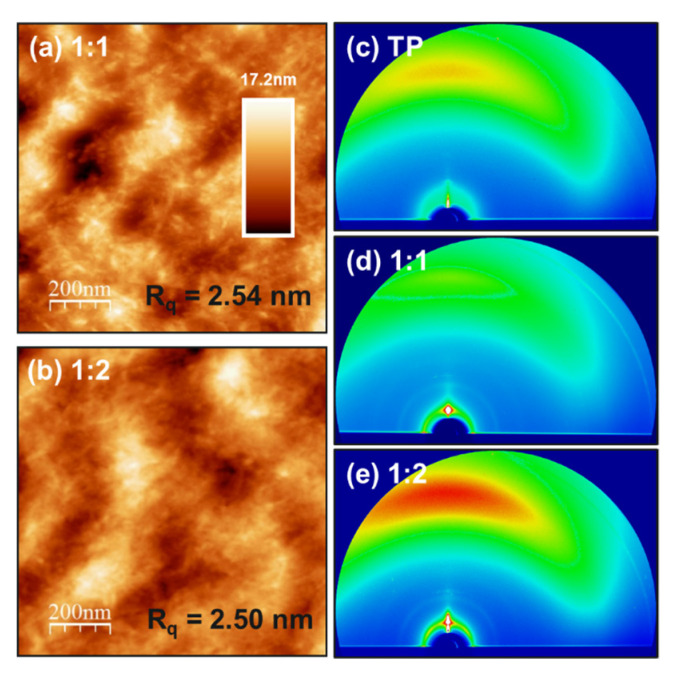
AFM images (1 µm × 1 µm) of PffBT4T-2OD:TP blends films of (**a**) 1:1 and (**b**) 1:2 weight ratio in tapping mode. 2D GIWAXS pattern of (**c**) neat TP, (**d**) 1:1, and (**e**) 1:2 blend films. (R_q_ is the root mean square roughness).

**Figure 3 materials-16-00737-f003:**
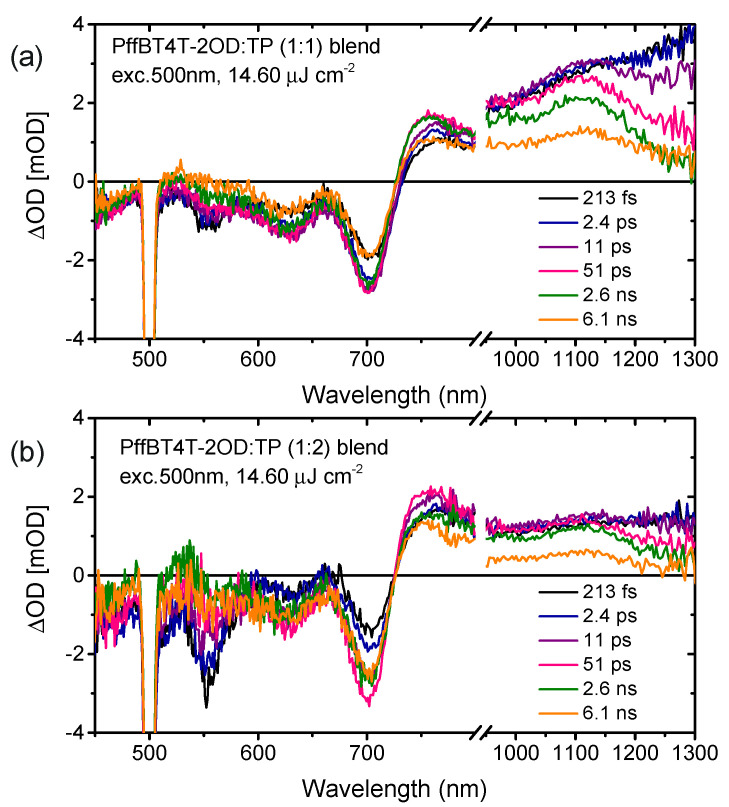
fs-Transient absorption spectra of blends of PffBT4T-2OD:TP of (**a**) 1:1 and (**b**) 1:2 weight ratio excited at 500 nm.

**Figure 4 materials-16-00737-f004:**
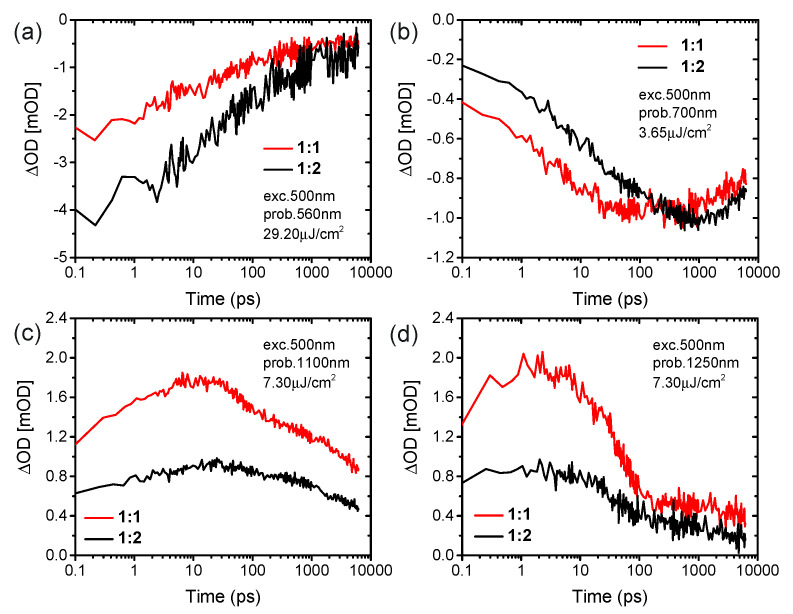
fs-Transient absorption dynamics at selected probe of blends of PffBT4T-2OD:TP of 1:1 and 1:2 weight ratio excited at 500 nm. (**a**) 560 nm, (**b**) 700 nm, (**c**) 1100 nm and (**d**) 1250 nm.

**Table 1 materials-16-00737-t001:** Device characteristics of PffBT4T-2OD:TP blends.

Materials	*J*_SC_ [mA cm^−2^]	*V*_OC_ [V]	*FF*	*PCE* [%]
PffBT4T-2OD:TP (1:1)	13.09	0.84	0.54	6.29
PffBT4T-2OD:TP (1:2)	12.78	0.82	0.54	5.61

## Data Availability

Not applicable.

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
