# Peer review of "Efficient Hole Transfer from a Twisted Perylenediimide Acceptor to a Conjugated Polymer in Organic Bulk-Heterojunction Solar Cells"

_materials, 2023, doi:10.3390/ma16020737_

Round 1
Reviewer 1 Report
This work studied the photovoltaic performance and hole transfer process in organic solar cells with a non-planar perylene dimer acceptor. fast exciton dissociation process from TP molecule to donor polymer was found by using femtosecond transient absorption spectroscopy (fs-TAS). This work is recommended to accepted after the following issues are addressed.
1. Only one donor:acceptor system was investigate in the manuscript, did the author tried other high-performance donors?
2. In the abstract, the singular and plural are not consistent in “a non-planar perylene dimer (TP) as electron acceptors”.
3. In page 4, “These devices show broad and strong photo response from 400 to 800 nm in entire EQE spectra”. The EQE is not “strong” compare with the recently reported OSCs. In addition, the EQE around 800 nm is very low. Better revise this sentence.
4. Some review paper about nonfullerene acceptors should be cited to facilitate the readers: J. Semicond., 2021, 42(10): 101607; J. Semicond, 2022, 43(5): 050203.
Author Response
(Reviewer’s comment) 1. Only one donor:acceptor system was investigate in the manuscript, did the author tried other high-performance donors?
(Author’s answers) We thank the reviewer for comment. Another donor:acceptor system has been published by Ravichandran Shivanna and coworkers (Energy Environ. Sci., 2014, 7, 435-441). In the paper, PBDTTT-CT:TP blends showed power conversion efficiency (PCE) of 3.2%, which is a half of PCE compared to the result (6.29%) in this work, because both PBDTTT-CT is more amorphous than PffBT4T-2OD.
(Reviewer’s comment) 2. In the abstract, the singular and plural are not consistent in “a non-planar perylene dimer (TP) as electron acceptors”.
(Author’s answers) According to the reviewer’s comment, we revised the phase, “a non-planar perylene dimer (TP) as an electron acceptor” (in Line 13)
(Reviewer’s comment) 3. In page 4, “These devices show broad and strong photo response from 400 to 800 nm in entire EQE spectra”. The EQE is not “strong” compare with the recently reported OSCs. In addition, the EQE around 800 nm is very low. Better revise this sentence.
(Author’s answers) According to the reviewer’s comment, we revised the sentence, “These devices show broad and strong photo response from 400 to 700 nm in entire EQE spectra” (in Line 138)
(Reviewer’s comment) 4. Some review paper about nonfullerene acceptors should be cited to facilitate the readers: J. Semicond., 2021, 42(10): 101607; J. Semicond, 2022, 43(5): 050203.
(Author’s answers) According to the reviewer’s comment, we added two references;
- Ye, L.; Ye, W.; Zhang, S. Recent advances and prospects of asymmetric non-fullerene small molecule acceptors for polymer solar cells. Semicond. 2021, 42, 101607.
- Ji, Y.; Bai, H.; Zhang, L.; Zhang, Y.; Ding, L. Nonfullerene acceptors based on perylene monoimides. Semicond, 2022, 43, 050203.

Reviewer 2 Report
In this paper, fullerene-free BHJ solar cells incorporating a TP 62 as an electron acceptor achieved PCEs of 6.29%, with high JSC of 13.94 mA cm-2 and VOC of 63 0.84 V were we demonstrated, compared the nanoscales morphologies of PffBT4T-2OD:TP blend films at differ- 64 ent composition ratios, and correlated these with exciton generation, exciton dissociation 65 and charge carrier recombination dynamics. This paper is interesting and can be recommended for publication in “Materials” after minor revisions.
1. Fig.2: SEM photographs should be more visualized than that of AFM;
2. Steady state photoluminescence and absorption spectra should be provided in the manuscript;
3. The noise of the transient absorption spectra was comparable to the signal variation, it should be reduced.
Author Response
(Reviewer’s comment) 1. Fig.2: SEM photographs should be more visualized than that of AFM;
(Author’s answers) We thank the reviewer for comment. Although PBDTTT-CT:TP blends showed comparable roughness in AFM, they do not have any crystallinity in the GIWAXS data. We believe GIWAXS data is stronger than SEM images for understanding material crystallinity and aggregation.
(Reviewer’s comment) 2. Steady state photoluminescence and absorption spectra should be provided in the manuscript;
(Author’s answers) We thank the reviewer for comment. We have already added steady state UV absorption in Figure 1(d) and photoluminescence in Figure S1.
(Reviewer’s comment) 3. The noise of the transient absorption spectra was comparable to the signal variation, it should be reduce
(Author’s answers) Thank you for your valuable comment. However, we are afraid of the misreading after smoothing of transient absorption spectra. The data has already shown the clear tendency of photoinduced absorption kinetics.

Reviewer 3 Report
The author reported an example of using TP as electron acceptors and achieve a power conversion efficiency of 6.29 % in a fullerene free solar cell. I have a few comments and questions:
• Can the author mentioned PCE is the abbreviation of power conversion efficiency? (line 63)?
• What is the full name of FF (line 47)?
• In page 5, the author mentioned "The GIWAXS two-dimensional (2D) maps of the pure TP and 177 both PffBT4T-2OD:TP blend films are shown in Figure 2(c)-(e)." Can the author add more detailed description of the shape and color of the GIWAXS 2D map?
• Rq was labeled in Figure 2(a)-(b). Is that average root mean square roughness?
• How did the author obtain FF and PCE? Any equation, calculation or measurement?
Author Response
(Reviewer’s comment) • Can the author mentioned PCE is the abbreviation of power conversion efficiency? (line 63)?
(Author’s answers) According to the reviewer’s comment, we added “… and have achieved power conversion efficiencies (PCEs) of over 18%” (in Line 26)
(Reviewer’s comment) • What is the full name of FF (line 47)?
(Author’s answers) According to the reviewer’s comment, we added “… and a fill factor (FF) of 0.46.” (in Line 46)
(Reviewer’s comment) • In page 5, the author mentioned "The GIWAXS two-dimensional (2D) maps of the pure TP and 177 both PffBT4T-2OD:TP blend films are shown in Figure 2(c)-(e)." Can the author add more detailed description of the shape and color of the GIWAXS 2D map?
(Author’s answers) According to the reviewer’s comment, we added “We note that the change from blue to red represents the improvement.” (in Line 180)
(Reviewer’s comment) • Rq was labeled in Figure 2(a)-(b). Is that average root mean square roughness?
(Author’s answers) According to the reviewer’s comment, we added the sentence in Figure 2, “(Rq is the root mean square roughness)” (in Line 166)
(Reviewer’s comment) • How did the author obtain FF and PCE? Any equation, calculation or measurement?
(Author’s answers) FF is observable parameter as well as JSC and VOC, although PCE is obtained as just a fraction of JSC × VOC × FF.
